# Structure of χ_3_-Borophene Studied by Total-Reflection High-Energy Positron Diffraction (TRHEPD)

**DOI:** 10.3390/molecules27134219

**Published:** 2022-06-30

**Authors:** Yuki Tsujikawa, Makoto Shoji, Masashi Hamada, Tomoya Takeda, Izumi Mochizuki, Toshio Hyodo, Iwao Matsuda, Akari Takayama

**Affiliations:** 1Institute for Solid State Physics, The University of Tokyo, Chiba 277-8581, Japan; imatsuda@issp.u-tokyo.ac.jp; 2Department of Physics and Applied Physics, Waseda University, Tokyo 169-8555, Japan; m.shoji-bmae2358@ruri.waseda.jp (M.S.); hamcchi.8@akane.waseda.jp (M.H.); zuratantakeabc@fuji.waseda.jp (T.T.); 3Institute of Materials Structure Science, High Energy Accelerator Research Organization (KEK), Ibaraki 305-0801, Japan; mochizu@post.kek.jp (I.M.); hyodot@post.kek.jp (T.H.)

**Keywords:** monolayer material, borophene, diffraction, TRHEPD

## Abstract

We have investigated the structure of χ_3_-borophene on Ag(111), a monolayer material of boron atoms, via total-reflection high-energy positron diffraction (TRHEPD). By comparing the experimental rocking-curves with ones for several structures calculated by using dynamical diffraction theory, we confirmed that the χ_3_-borophene layer has a flat structure. The distance from the topmost layer of the metal crystal is 2.4 Å, which is consistent with results reported by X-ray standing wave-excited X-ray photoelectron spectroscopy. We also demonstrated that the in-plane structure of χ_3_-borophene is compatible with the theoretical predictions. These structural properties indicate that χ_3_-borophene belongs to a group of epitaxial monolayer sheets, such as graphene, which have weak interactions with the substrates.

## 1. Introduction

Monolayer (ML) materials that are composed of a single element, which are referred to by the term “Xene”, have attracted great interest. It is because they have intriguing electronic states, such as Dirac Fermions [1], which are derived by a structural arrangement of a two-dimensional (2D) honeycomb lattice. Today, Xenes have been investigated by a large number of researchers in both the academic and technological fields. Typical examples of Xenes are composed of group 14 elements and include materials such as graphene, silicene, and germane [2,3]. Recently, a 2D material of boron (a group 13 element) called “borophene” has been attracting attention, especially after the success of its fabrication on metal substrates [4,5,6,7,8]. Similar to 3D boron, borophene layers are predicted to have various types of atomic structures due to the multi-center bonding scheme of boron atoms [9]. Theoretical works have found that a flat layer of boron is stable when it is composed of triangular lattices and hexagonal hollows [9,10,11]. Experimental observations of epitaxial borophene on Ag(111) have been consistent with the theoretical structural models [6,12], but the atomic structure has not been examined directly. Thus, there is a strong need to conduct structural analysis by an appropriate diffraction experiment on the surface.

In the present research, we studied the atomic structure of χ_3_-borophene on Ag(111) by means of total-reflection high-energy positron diffraction (TRHEPD). This structural analysis method is exceedingly surface-sensitive, and it has determined varieties of atomic layers precisely [1,13]. The present experiment has revealed that *χ*_3_-borophene is flat in the 2D layer with a distance of 2.4 Å from the Ag(111) surface. This result agrees with the structural models proposed in previous experimental and theoretical studies [12,14]. Notably, its structure is more like graphene, which interacts weakly with the substrate, and unlike silicene and germane, which are epitaxial on the metal substrate.

## 2. Results and Discussion

### 2.1. Calculated Rocking Curves

Since this work is the first case of applying TRHEPD for structural analysis of a 2D boron material, here we briefly introduce the principles and the methods (see Section 3.2 for details). The TRHEPD experiment consists of measuring a series of the diffraction patterns for a fixed incident azimuthal direction at various glancing angles. Following that, the diffraction intensity of the specular (00) spot is plotted as a function of *θ*. The plot is called a rocking curve (RC). In the structural analysis, the experimental RCs are compared with those calculated for various structural models by using dynamical diffraction theory. A reliability factor R [15,16] is used as a criterion to judge goodness of the agreement. Typical measurements are made under one-beam (OB) and many-beam (MB) conditions. In the OB condition the beam is incident along an off-symmetric direction, while in the MB condition it is incident along a symmetric direction. The RC in the OB condition essentially gives the information on the atomic positions in the out-of-plane direction only, while the RC in the MB condition includes information on the in-plane structure as well [17,18]. Before showing the experimental data of the TRHEPD and the optimized results, we demonstrate the sensitivity of the RC of TRHEPD to the details of structural models. Figure 1a shows the top view of a structural model of $\chi_{3}$-borophene on Ag(111), with arrows indicating the beam incident directions for the OB (red) and MB (blue) conditions. The positron beam is incident with a 17° deviation from the [1$\bar{1}$0] direction for the OB condition, while it is along the [1$\bar{1}$0] direction for the MB condition. Usually, the TRHEPD analysis of the data under the OB condition is performed first to extract the out-of-plane structure only, i.e., in the present case, the interlayer distance between the Ag substrate and borophene and the possible buckling of the borophene sheet.

#### 2.1.1. One-Beam (OB) Condition

Figure 2b,c show the calculated RCs under the OB condition using the structural model of the $\chi_{3}$-borophene [12]. The interlayer distance *d* from the substrate and the magnitude of the buckling Δ *i*n the layer were changed systematically. It is of note that the proposed model [12] has the parameters *d* = 2.4 Å and Δ = 0.04 Å. The calculations here assume that the 1 ML borophene sheet completely covers the Ag substrate. Figure 1b shows RCs calculated for various values of *d* with Δ fixed to 0.0 Å (i.e., flat or no buckling). One can find that the shape of the RC apparently depends on *d*. For example, a peak near 4° at *d* = 2.0 Å approaches 3° at *d* = 3.0 Å. Figure 1c shows a series of RCs calculated for various Δ with *d* fixed at 2.2 Å. Furthermore, it is clearly seen that the shape of the RC changes as Δ increases. These results in Figure 1b,c demonstrate that RCs measured by TRHEPD are sensitive enough to accurately examine the *d* and Δ values of borophene.

#### 2.1.2. Many-Beam (MB) Condition

The in-plane structure is determined from the RCs under MB condition; for χ_3_-borophene on the Ag(111), the beam is incident along the [1$1\bar{2}$] and [1$\bar{1}$0] directions. It is reported that the χ_3_-borophene on the Ag(111) has three domains that are 120°-rotated from each other [14]. The structures in the different domains are illustrated in Figure 2a–c. Essentially, we must pay attention to the domain direction relative to the beam incidence direction. Interestingly, the calculated results of RCs of the three domains along the [1$\bar{1}$0] direction are identical to one another, as presented in Figure 2d–f. This is because the in-plane structure perpendicular to the beam incidence direction is reflected in the RC. It means that existence of domains is not an issue when the incident positron beam is along the [1$\bar{1}$0] direction. In this study, we focus on data along the [1$\bar{1}$0] direction for a sample without domain control.

### 2.2. TRHEPD Measurements

#### 2.2.1. OB Measurement

Figure 3 shows the RCs of (a) a pristine Ag(111) surface and (b) the borophene grown on it, measured under the OB condition. In Figure 3a, the experimental curve is superimposed on the calculated one, based on a previous TRHEPD report of Ag(111) [2]. The profiles show good agreement with each other except for a small deviation below 2°. The suppressed intensity at the low glancing angles is likely due to the surface roughness, or terraces with various heights on the Ag substrate. It has been made from the multiple sputtering and annealing processes breaking the ideal total-reflection condition. Such an effect is observed only in TRHEPD measurement whose surface selectivity is exceedingly high and is not a concern in other techniques of surface analysis such as electron diffraction. Considering this, we only used data above 1.6° for accurate analysis.

In Figure 3b, a result for the $\chi_{3}$-borophene layer grown on the Ag(111) substrate obtained in the OB condition was compared with the corresponding calculation. The experimental profile was reproduced by the proposed model with *d* = 2.4 Å and Δ = 0.04 Å [12]. In addition, it resulted in a surface occupancy of 56% on the Ag(111) substrate with a minimum R of 1.38%. These agree with the previous report of the microscopic imaging by scanning tunneling microscopy that observed a surface covered partially by domains of the borophene [6]. We found that the agreements between the experiment and the calculation were never possible with a structure with appreciable buckling. This indicates that there is no buckling considering by the sensitivity of the shape of the RC to the amount of buckling as shown in Figure 1c. Using the diffraction method, we have directly confirmed that the $\chi_{3}$-borophene layer is a flat sheet and found that the structural model determined in this study is completely consistent with that previously proposed by the X-ray standing wave photoemission experiment [12].

#### 2.2.2. MB Measurement

Figure 4 shows the RCs acquired by the THREPD measurements along the [1$\bar{1}$0] direction, i.e., under the MB condition, for (a) the pristine Ag(111) surface and (b) epitaxial $\chi_{3}$-borophene. As in the case of the data under the OB condition, the intensity at low glancing angles (<~2°) was suppressed due to surface roughness that partly hindered the total reflection of the positron beam. RCs measured under MB conditions include information on the in-plane and out-of-plane structure, i.e., the analysis must be conducted involving many parameters that are not independent. In this study, we have assumed the in-plane structural model proposed in previous theoretical studies as shown in Figure 1a [14], and adjusted *d* and Δ so that *R* for the glancing angles above 2.0° is smallest. We found that the experimental curves were well reproduced with the same parameters optimized for the OB condition, *d* = 2.4 Å, Δ = 0.0 Å, and 54% coverage. Other possible structural models would not reproduce the experimental RC well. Thus, the result supports the structural model proposed in theoretical studies, which state that $\chi_{3}$-borophene on Ag(111) is a structure with a triangular lattice and hexagonal hollows. In the present study we did not perform a full analysis of the MB RCs, where all the in-plane atomic coordinates are optimized. For a more accurate structure determination, we plan to prepare a sample with a single-domain and analyze the RC under MB conditions along the [1$1\bar{2}$] direction.

It would be worth while to discuss if or how TRHEPD is able to observe the concentric superlattice structure in borophene via self-assembly of twin boundaries. Previous STM works [12] have established the very interesting defect-mediated self-assembly as a pathway to unique borophene structures and properties. Unfortunately, considering the positron beam diameter of the current TRHPED measurement, it is difficult to observe and analyze such structures, which depend on local domains or boundaries. Structural analysis for these concentric superlattice structures may become possible if diffraction patterns originating from the structural periodicity of such domains could be observed, or by achieving microscopic measurement comparable to the domain size.

## 3. Materials and Methods

### 3.1. Sample

A clean surface of Ag(111) crystal was prepared by several cycles of Ar+ sputtering at 0.5 keV for 10 min and annealing processes at 450 °C, followed by confirmation through clear 1 × 1 pattern of reflection high-energy electron diffraction (RHEED). Subsequently, the epitaxial $\chi_{3}$-borophene was prepared by boron deposition at 300 °C. We checked the formation of the borophene layer with the 3 × 3 pattern by RHEED and TRHEPD, as shown in Figure 5a,b, respectively.

### 3.2. TRHEPD Method

#### 3.2.1. Features of TRHEPD

Figure 6 schematically shows a measurement configuration of the TRHEPD method. The set-up resembles that of RHEED that uses the electron, its antiparticle. The surface sensitivity of TRHEPD and RHEED results from the fact that elastic scattering or diffraction events can only be available from a surface within a depth of the inelastic mean free path. In addition, compared to electrons, a positron beam has the advantage of much higher surface sensitivity because it senses positive electrostatic potential in every material [15,19,20]; when a positron of kinetic energy $E_{0}$ is incident on a solid whose mean internal potential is $V_{0}$, the conservation of total energy leads to the mean kinetic energy (E) inside to be $E=E_{0}-eV_{0}$, while it is $E=E_{0}+eV_{0}$ in the case of an electron. Here, for incidence at a glancing angle (θ), the component of E related to the perpendicular momentum component inside materials ($E_{⊥}$) is described as $E_{⊥}=E_{0}\sin^{2}\theta +eV_{0}$ due to conservation of the parallel component of the momentum. Since no state is allowed for $E_{⊥}<0$, positrons are totally reflected for θ below the critical angle ($\theta_{C}$)[$\theta <\sin^{-1}\sqrt{eV_{0}/E_{0}}=\theta_{c}]$, while the positron beam penetrates into the crystal at the glancing angle above $\theta_{c}$.

#### 3.2.2. TRHEPD Analysis

The R-factor is defined as $R=\sqrt{\underset{i}{\sum }{\left[I_{\exp}\left(\theta_{i}\right)-I_{\mathrm{cal}}\left(\theta_{i}\right)\right]}^{2}}\times 100\;\left(\%\right)$ where $I_{\exp}\left(\theta_{i}\right)$ and $I_{\mathrm{cal}}\left(\theta_{i}\right)$ are the experimental and calculated intensities of a diffraction spot at the glancing angle $\theta_{i}$, respectively, normalized under the following condition: $\underset{i}{\sum }I_{\exp}\left(\theta_{i}\right)=\underset{i}{\sum }I_{\mathrm{cal}}\left(\theta_{i}\right)=1$. The most appropriate structure should be the one with the minimum R, as reduction in R corresponds to a smaller difference between the measured and the calculated RCs with structural optimization.

#### 3.2.3. Experimental Condition for TRHEPD

The experiment was performed at the Slow Positron Facility (SPF) in IMSS, KEK. The incident energy and the spot size of the positron beam were 10 keV and ϕ 1 mm, respectively. The TRHED rocking curves were measured in a glancing angle range of 0.3°~6.5° with 0.1° steps. All of the measurements were performed at room temperature. At incident energy *E* = 10 keV, the critical angle for an Ag crystal with $eV_{0}=23$ eV is $\theta_{C}$ = 2.8° [2]. Calculations of the rocking curves for TRHEPD were performed by the structure-analysis program, “2DMAT” [21], as well as the one build by Hanada et al. [22,23].

## 4. Conclusions

We performed a THRPED study to examine the atomic structure of the a $\chi_{3}$-borophene layer on Ag(111). The structural parameters with *d* = 2.4 Å and *Δ* = 0.04 Å, indicate that the $\chi_{3}$-borophene belongs to a group of the monolayer sheets that have weak interactions with substrates, such as graphene. The analysis of the RC under the MB condition along the [1$\bar{1}$0] direction was also compatible with the in-plane structural model proposed by theoretical structural model for $\chi_{3}$-borophene. The present study is the first application of the diffraction method for the structural analysis of borophene. The method of THRPED is described in some detail because the techniques is novel, and it can be applied to the other 2D boron materials.

## Figures and Tables

**Figure 1 molecules-27-04219-f001:**
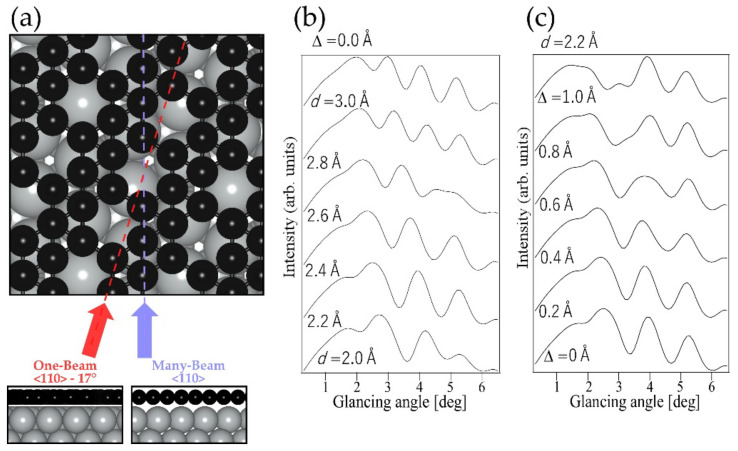
(**a**) Structural model of $\chi_{3}$-borophene. Beam directions of the one- and many-beam condition are depicted by the red and blue arrows, respectively. The bottom pictures show the side view along each condition. (**b**) Calculated rocking curves of $\chi_{3}$ -borophene in one-beam condition with different interlayer distance *d*, with no buckling (Δ = 0.0Å ). (**c**) Calculated rocking curves of $\chi_{3}$ -borophene with different buckling Δ, with fixed interlayer distance (*d* = 2.2 Å) which is the height of the lowest boron atoms from the Ag substrate.

**Figure 2 molecules-27-04219-f002:**
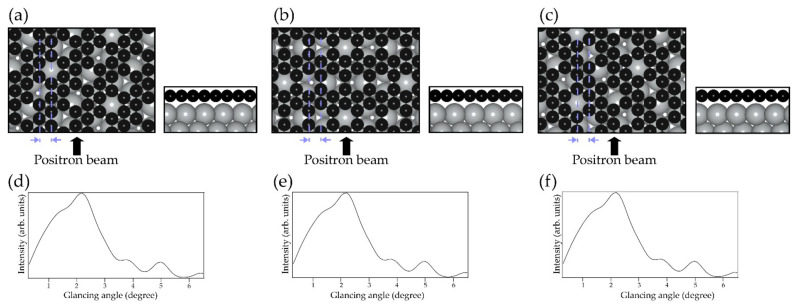
(**a**–**c**) Structure of each of the three domains of χ_3_-borophene grown on different directions to Ag(111), viewed from the top and side. (**d**–**f**) The RCs of (**a**–**c**), respectively. The shape of the RCs was the same for all.

**Figure 3 molecules-27-04219-f003:**
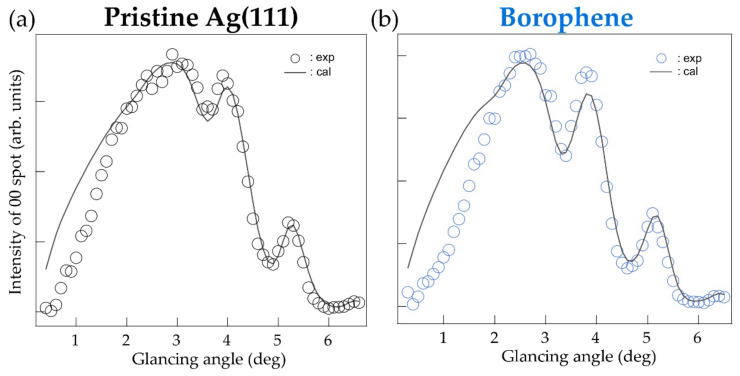
Rocking curves under one-beam (OB) condition with calculated curves for (**a**) pristine Ag(111) and (**b**) χ_3_-borophene on Ag(111).

**Figure 4 molecules-27-04219-f004:**
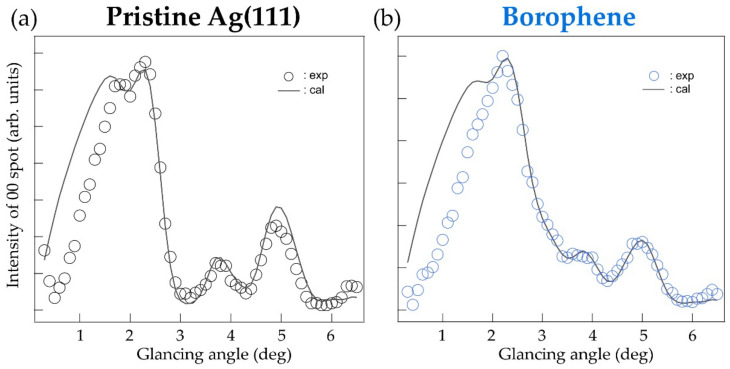
Rocking curve under many-beam (MB) condition with calculated curves for (**a**) pristine Ag(111) and (**b**) $\chi_{3}$-borophene on Ag(111).

**Figure 5 molecules-27-04219-f005:**
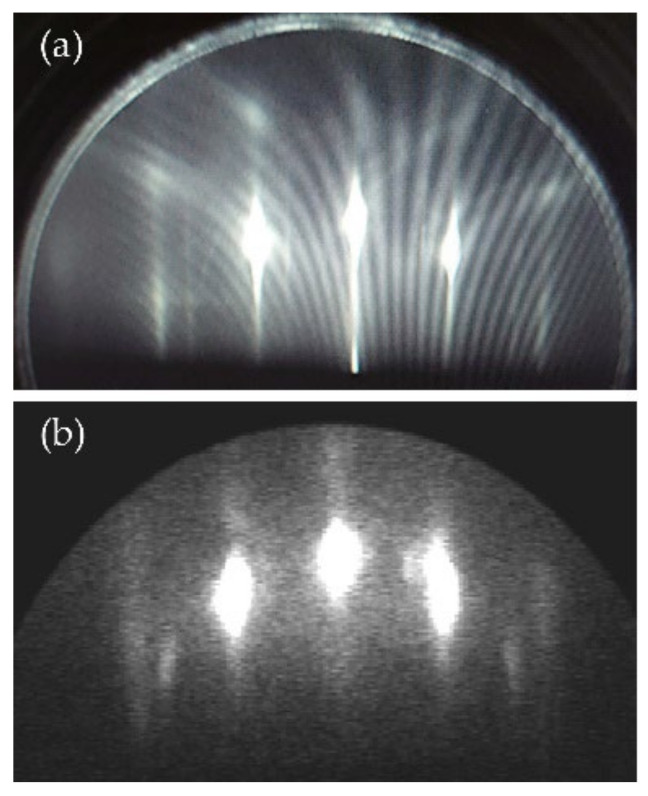
(**a**) The RHEED and (**b**) TRHEPD pattern of the epitaxial $\chi_{3}$-borophene on Ag(111).

**Figure 6 molecules-27-04219-f006:**
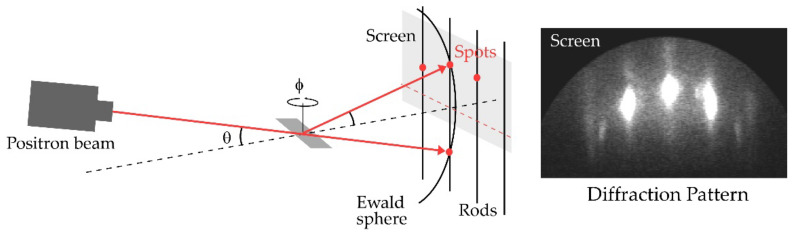
Schematic diagram of TRHEPD. A positron beam is incident on a sample surface with a glancing angle (θ) and an orientation fixed at a specific azimuthal angle (ϕ ). The diffraction spots appear at the intersection of the Ewald sphere and reciprocal-lattice rods and the spots are projected onto the screen as the diffraction pattern on the right.

## Data Availability

Data is contained in article.

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
