# Peer review of "Structure of χ_3_-Borophene Studied by Total-Reflection High-Energy Positron Diffraction (TRHEPD)"

_molecules, 2022, doi:10.3390/molecules27134219_

Round 1
Reviewer 1 Report
In this manuscript, Tsujikawa et al, presented a high-energy positron diffraction investigation of ?3-borophene structure and finds it to be a member of monolayer family. The manuscript is overall written good and nicely arranged. I recommend for publication, however few minor pointes need to include:
1. Fig.d is missing. 2. Good quality surface preparation of Ag(111) is important. So authors can include the energy and time for each Ar+ sputtering process.
Author Response
In this manuscript, Tsujikawa et al, presented a high-energy positron diffraction investigation of χ3-borophene structure and finds it to be a member of monolayer family. The manuscript is overall written good and nicely arranged. I recommend for publication, however few minor pointes need to include:
Thank you very much for recognizing our work and suggestions for improving the manuscript. Below we address the comments.
1. Fig.d is missing.
In the caption of Figure 1, (d) was confusing. To address this consistently we have removed the parentheses of (d) and (Δ) everywhere.
2. Good quality surface preparation of Ag(111) is important. So authors can include the energy and time for each Ar+ sputtering process.
The reviewer's point is well taken. We have added information about sputtering energy and time (Page 5 lines 181-182).
Reviewer 2 Report
This paper from Tsujikawa and collaborators which experimentally reports on the structure of ??- borophene is an ok work which seems to have been competently carried out. The structure of borophene is already known based on theory and experiments (see ref. 14 for instance). The interest of this paper resides in the fact that it is confirmed by total-reflection high-energy positron diffraction (TRHEPD) technique for the first time. I guess the works merits to be published. However, I am not sure that Molecules is the best medium for that. Maybe, a journal devoted to materials science would be more appropriate.
Concerning the contents of the paper, I have just a couple of minor comments which should be addressed before publication proceeds.
a) Introduction. Latin numbers for groups of elements in the Periodic Table are not used anymore. Arabic numbers are recommended by IUPAC.
b) Previous works have established defect-mediated self-assembly as a pathway to unique borophene structures and properties (see ref. 14 for instance). This is not discussed here. In other words, is the TRHEPD technique able to observe this phenomenon?
c) Minor points (typos)
Line 18. which is consistent with results reported by X-ray standing wave- 18 excited X-ray photoelectron spectroscopy
Line 21. a group of the epitaxial monolayer sheets,
Line 228. The method of THRPED is described in the article since the technique is novel and can be applied to the other 2D boron materials.
Author Response
This paper from Tsujikawa and collaborators which experimentally reports on the structure of χ3- borophene is an ok work which seems to have been competently carried out. The structure of borophene is already known based on theory and experiments (see ref. 14 for instance). The interest of this paper resides in the fact that it is confirmed by total-reflection high-energy positron diffraction (TRHEPD) technique for the first time. I guess the works merits to be published. However, I am not sure that Molecules is the best medium for that. Maybe, a journal devoted to materials science would be more appropriate. Concerning the contents of the paper, I have just a couple of minor comments which should be addressed before publication proceeds.
We appreciate Reviewer’s suggestion to submit this paper to a journal devoted to materials science. But, this time, we would like to inform as many boron researchers as possible about the new THRPED technique and our work on the structure of borophene determined by THRPED. So we decided to submit our work to a special issue “New Science of Boron Allotropes, Compounds, and Nanomaterials” in Molecules. And thank you very much for suggestions of improving the manuscript. Below we address the comments.
a) Introduction. Latin numbers for groups of elements in the Periodic Table are not used anymore. Arabic numbers are recommended by IUPAC.
We have revised the manuscript as so.
b) Previous works have established defect-mediated self-assembly as a pathway to unique borophene structures and properties (see ref. 14 for instance). This is not discussed here. In other words, is the TRHEPD technique able to observe this phenomenon?
As the reviewer points out, the existence of concentric superlattices via self-assembly of twin boundaries in borophene on Ag(111) has been reported in STM measurements. Such structures are very interesting. However, considering the spatial resolution of the current TRHPED measurement, it is difficult to observe and analyze such structures that depend on local domains or boundaries. Structural analysis may become possible if diffraction patterns originating from the structural periodicity of such domains could be observed (unfortunately not observed in this experiment), or by achieving microscopic measurement comparable to the domain size. We add these explanations to the revised manuscript (Page 5 lines 165-173).
c) Minor points (typos)
Line 18. which is consistent with results reported by X-ray standing wave- 18 excited X-ray photoelectron spectroscopy
Line 21. a group of the epitaxial monolayer sheets,
Line 228. The method of THRPED is described in the article since the technique is novel and can be applied to the other 2D boron materials.
Thank you for pointing these out. We have corrected these typos.